# Persistence of intact HIV-1 proviruses in the brain during antiretroviral therapy

Weiwei Sun[1], Yelizaveta Rassadkina[1], Ce Gao[1], Sarah Isabel Collens[2], Xiaodong Lian[1], Isaac H Solomon[3], Shibani S Mukerji[2], Xu G Yu[1,4], Mathias Lichterfeld[1,4]*

[1]Ragon Institute of MGH, MIT and Harvard, Cambridge, United States; [2]Department of Neurology, Massachusetts General Hospital, Boston, United States; [3]Department of Pathology, Brigham and Women's Hospital, Boston, United States; [4]Infectious Disease Division, Brigham and Women's Hospital, Boston, United States

**Abstract** HIV-1 reservoir cells that circulate in peripheral blood during suppressive antiretroviral therapy (ART) have been well characterized, but little is known about the dissemination of HIV-1-infected cells across multiple anatomical tissues, especially the CNS. Here, we performed single-genome, near full-length HIV-1 next-generation sequencing to evaluate the proviral landscape in distinct anatomical compartments, including multiple CNS tissues, from 3 ART-treated participants at autopsy. While lymph nodes and, to a lesser extent, gastrointestinal and genitourinary tissues represented tissue hotspots for the persistence of intact proviruses, we also observed intact proviruses in CNS tissue sections, particularly in the basal ganglia. Multi-compartment dissemination of clonal intact and defective proviral sequences occurred across multiple anatomical tissues, including the CNS, and evidence for the clonal proliferation of HIV-1-infected cells was found in the basal ganglia, in the frontal lobe, in the thalamus and in periventricular white matter. Deep analysis of HIV-1 reservoirs in distinct tissues will be informative for advancing HIV-1 cure strategies.

**\*For correspondence:** mlichterfeld@mgh.harvard.edu

**Competing interest:** The authors declare that no competing interests exist.

**Sent for Review** 07 June 2023

**Preprint posted** 26 June 2023

**Reviewed preprint posted** 01 August 2023

**Reviewed preprint revised** 27 September 2023

**Version of Record published** 08 November 2023

## eLife assessment

This **important** study uses near full-length HIV-1 sequencing to examine proviral persistence in various tissues derived from three individuals who received antiretroviral therapy until time of death. Intact as well as defective HIV-1 proviruses are found at various anatomical sites including the central nervous system; the results are **convincing** and relevant for our understanding of latent viral reservoirs, especially in the brain.

## Introduction

Combination antiretroviral therapy has extended the life expectancy of people living with HIV (PLWH) to near-normal levels, but it is clear that antiretroviral therapy (ART), in its present form, does not lead to a sustained, drug-free remission of HIV-1 infection; instead, a long-lived reservoir of virally-infected cells persists despite ART (*Chun et al., 1997*; *Wong et al., 1997*) and can effectively fuel rebound viremia in case of treatment interruptions. Hence, ART needs to be taken life-long; finding a way to eliminate HIV-1 reservoir cells remains critical to finding a cure for HIV-1 infection.

HIV-1 reservoir cells that circulate in the peripheral blood have been well-described in recent years (*Margolis et al., 2020*). These circulating reservoir cells mostly consist of memory CD4 T cells that persist life-long and harbor chromosomally-integrated viral DNA, also referred to as a 'provirus'. The pool size of HIV-1-infected cells during suppressive antiretroviral therapy is frequently maintained or replenished by clonal proliferation of viral reservoir cells, during which an identical viral DNA copy

**eLife digest** Approximately 39 million people in the world live with HIV infection. Currently available treatments can reduce the amount of virus to near undetectable levels. But they do not eliminate the virus. A reservoir of HIV-infected cells persists during treatment. If treatment stops, these cells can cause rebounding virus levels and a return of symptoms. As a result, patients living with HIV must remain on treatment their entire lives.

HIV reservoir cells often do not express viral proteins, making them hard for the immune system to find and destroy. Many of these reservoir cells occur in lymph nodes, which makes them difficult for researchers to access for study. Learning more about where these cells hide in the body may enable scientists to develop new treatments to help eliminate them.

Sun et al. show that HIV reservoir cells exist in many body tissues, including the brain. In the experiments, Sun et al. used single HIV genome sequencing to identify HIV genetic sequences in the brain and other body tissues from three recently deceased individuals with HIV. The individuals agreed to donate their tissues for postmortem studies before their deaths. All received antiretroviral therapy until death. The experiments identified functional HIV genetic sequences in lymph nodes and gastrointestinal tissues, known hotspots for HIV-infected cells. Sun et al. also found genetically intact HIV in brain tissue from two of the individuals. The HIV genetic sequences were identical to sequences found in other body tissues. This discovery suggests HIV-infected cells had divided into more HIV-infected cells and spread.

The results suggest that cells harboring intact HIV invade the brain and persist there for extended periods during antiretroviral therapy. To eradicate the virus, interventions targeting HIV reservoir cells must be able to reach the brain. This new information may help researchers developing HIV-reservoir targeting drugs decide which candidates will likely be the most effective. Future studies may also shed light on how HIV reaches the brain and how the infected cells escape destruction by immune cells, which may suggest more treatment strategies.

is passed on to daughter cells (*Cohn et al., 2015*; *Hosmane et al., 2017*; *Bui et al., 2017*; *Einkauf et al., 2019*; *Pinzone et al., 2019*). Important advances in recent technology development, including single-genome near full-length proviral sequencing, have demonstrated that the vast majority of HIV-1 DNA species encountered in ART-treated persons are defective (*Ho et al., 2013*; *Lee et al., 2017*; *Hiener et al., 2017*), mostly due to errors occurring during reverse transcription of viral RNA into DNA. Such errors can lead to large deletions, premature stop codons, or defects in the viral packaging signal region; moreover, specific host proteins such as APOBEC-3G/3 F can induce lethal hypermutations into the proviral sequence. The ability of viral reservoir cells to persist indefinitely is frequently attributed to very limited or absent proviral gene transcription; this viral latency protects infected cells from host immune responses and reduces possible cytopathic effects associated with viral gene expression. However, recent studies have emphasized that proviral gene expression can be ongoing in some HIV-1-infected cells during antiretroviral therapy (*Yukl et al., 2018*), typically when proviruses are integrated in immediate proximity to activating chromatin marks (*Einkauf et al., 2022*; *Lian et al., 2023*).

Much less is currently known about HIV-1 reservoir cells in other body compartments that are more difficult to access for analytic purposes. Lymph nodes and lymphoid tissues in the gastrointestinal tract harbor the vast majority of all lymphocytes and are likely to represent a prime location for the persistence of virally infected cells (*Estes et al., 2017*; *Baiyegunhi et al., 2022*; *Kroon et al., 2022*; *Beckford-Vera et al., 2022*); however, studies that interrogate viral reservoir cells in these tissues remain limited. Even less is known about HIV-1 persistence in other organs, and, in particular, in the CNS, although recent studies have started to explore viral reservoir cells in such specific body compartments (*Cochrane et al., 2022*). In the present study, we used single-genome proviral sequencing to conduct a detailed analysis of HIV-1 proviral sequences in autopsy samples from up to n=18 different organ systems from three study participants.

**Table 1.** Clinical and demographical data of study participants.

| Participant | Gender | Age of death | Duration of HIV-1 infection (yr) from diagnosis date | Time on HAART (yr) | HAART regimen | CD4 count before death (cells/ul) | Viral load before death (copies/ml) |
|---|---|---|---|---|---|---|---|
| 1 | Female | 38 | 16 | 16 | FTC, RPV, TAF | 1625 | Undetectable |
| 2 | Male | 68 | 25 | 1 | FTC, TAF, BIC | 165 | Undetectable |
| 3 | Male | 52 | 1 | 1 | BIC, FTC, TAF | 145 | 136 |

## Results

### Frequency of intact and defective proviruses in tissue compartments

To investigate the proviral landscape across multiple anatomical tissues, including the CNS, we focused on three participants from whom post-mortem tissue samples were available for HIV-1 research. Tissue samples from the CNS and other organs were collected by a rapid (<24 hr) autopsy after death. The clinical and demographic characteristics of these study participants are shown in *Table 1*. All study participants adhered to antiretroviral treatment until death; plasma viral loads were undetectable by commercial assays in study participants 1 and 2, in whom 14 and 15 different tissue sections were sampled, respectively (*Table 2*). Organ-specific tissues analyzed in these two study participants included lymph node, spleen, colon, liver, pancreas, kidney, thyroid gland, and adrenal gland; in the female study participant 1, ovarian and uterus tissues were analyzed, while in the male study participant 2, prostate and testicular tissues were studied. In both of these study participants, four different CNS tissue sections (basal ganglia, thalamus, frontal lobe, and occipital lobe) were

**Table 2.** Cell numbers analyzed from each tissue of each study participant.

| | Compartment | Participant 1 (million cells) | Participant 2 (million cells) | Participant 3 (million cells) |
|---|---|---|---|---|
| | Basal ganglia | 67.25 | 18.06 | 32.85 |
| | Thalamus | 35.81 | 5.77 | 21.83 |
| | Occipital lobe | 87.17 | 29.07 | 43.65 |
| | Frontal lobe | 64.29 | 13.39 | 64.91 |
| Brain tissues | Periventricular white matter | | | 36.42 |
| | Lymph nodes | 28.86 | 15.57 | |
| | Spleen | 63.69 | 104.15 | |
| | Colon | 54.02 | 17.74 | |
| | Liver | 72.91 | 43.59 | |
| | Pancreas | 91.36 | 65.1 | |
| | Terminal ileum | | 1.46 | |
| | Kidney | 57.02 | 25.9 | |
| | Ovary | 51.96 | | |
| | Uterus | 79.11 | | |
| | Testes | | 4.74 | |
| | Prostate | | 19.5 | |
| | Thyroid | 53.65 | 46.63 | |
| Non-brain tissues | Adrenal | 39.43 | 14.87 | |
| | Total | 846.53 | 425.54 | 199.66 |

collected for investigation. In study participant 3, plasma viral load was 136 copies/ml at the time of death; 5 different tissue sections from the CNS (basal ganglia, thalamus, occipital lobe, frontal lobe, periventricular white matter) were analyzed in this person.

Near full-length single-template next-generation HIV-1 proviral sequencing was performed to profile the proviral reservoir landscape at single-molecule resolution in tissue samples. The number of cells analyzed from each organ in each participant is listed in *Table 2*. In total, 846.53, 425.54, and 199.66 million cells were assayed in study participants 1, 2, and 3, respectively, resulting in 1471.73 million cells analyzed in all study participants combined. A total of 1497 individual proviral sequences were amplified, of which n=497 were selected for next-generation sequencing based on their amplicon sizes on gel electrophoresis; the remaining sequences were classified as proviruses with large deletions. All amplicons (n=74) from CNS tissues were sequenced, regardless of their length. Using a previously described computational pipeline to identify lethal defects in proviral sequences, we identified 48 proviruses (3.21% of all proviruses) that met our criteria for genome-intactness (*Figure 1A–C*); this number is consistent with the small number of genome-intact proviruses detected in previous studies. Many sequences, both genome-intact and defective, were identified multiple times, consistent with clonal proliferation of infected cells (*Figure 1B*), as reported in prior work (*Bui et al., 2017*; *Pinzone et al., 2019*; *Lee et al., 2017*; *Hiener et al., 2017*).

To evaluate HIV-1 persistence in selected tissue compartments, the frequencies of total, intact, and defective HIV-1 proviruses in each tissue were analyzed (*Figure 2A–C*). Intact proviral sequences were only detected in 8 tissue sites, including basal ganglia, periventricular white matter, lymph node, spleen, colon, kidney, prostate, and the thyroid gland (*Figure 2B*). The numbers of intact HIV-1 sequences in these eight tissues varied from 0.01 to 0.6 copies per million cells. Consistent with previous studies, the frequency of intact proviral sequences was highest in the lymph node in participant 1 (0.52 intact proviruses/million cells) and participant 2 (0.58 intact proviruses/million cells), followed by kidney, spleen, colon, and basal ganglia in participant 1 and by prostate, spleen, thyroid in participant 2. Intact proviruses were detected in the basal ganglia in study participant 1 (frequency of 0.015 /million cells) and in study participant 3 (0.030 intact proviruses/million cells). Moreover, one intact provirus was also detected in periventricular white matter in participant 3 (0.027 intact proviruses/million cells), in whom analysis was limited to brain tissues. No intact proviruses were detected in the CNS tissues of study participant 2, despite analyzing 66.29 million cells. Together, these results indicate that intact HIV-1 proviruses are preferentially detected in lymphoid and gastrointestinal (GI) tissues. The frequency of intact proviruses in the CNS is comparatively low; however, this study is the first one to document the presence of genome-intact proviral sequences in CNS tissues using near full-length proviral sequencing.

Defective proviral sequences were detected in all analyzed tissue samples except for the thalamus from participant 1 (*Figure 2C*). In participant 1, the frequency of defective proviral sequences was highest in lymph nodes (5.7 defective proviruses/million cells), followed by colon, spleen, and kidney. In participant 2, the frequency of defective proviral sequences was highest in lymph nodes (13.0 defective proviruses/million cells), followed by prostate, colon, and spleen. Notably, the prostate had a very high frequency of virally infected cells. The ratio of intact to defective proviral species was relatively high among sequences isolated from the basal ganglia of participants 1 and 3 (*Figure 2— figure supplement 1*). Taken together, these results demonstrate the highest frequencies of defective proviruses in lymph nodes, in the colon and in the prostate. CNS tissues contained relatively low frequencies of proviral sequences, compared to other tissue sites; however, defective proviruses were isolated in all but one of the analyzed 13 different CNS samples.

## Phylogenetic associations and clonality

A series of prior studies demonstrated large sequence-identical clones of intact and defective proviruses in the peripheral blood of ART-treated study participants. In our subsequent analysis, we studied the dissemination of clonal proviral sequences across different tissues. In participant 1, a total of 24 intact proviral sequences were detected. Two large clones of intact proviruses were observed, one of which included sequences detected in kidney, lymph node, and spleen samples (*Figure 3A–B*). The other clone involved intact proviral sequences from basal ganglia and lymph node tissues. Among 218 defective proviruses sequenced in participant 1, 14 clones were observed across multiple tissues (*Figure 3C*). In participant 2, among 22 intact proviral sequences, only one clone with two member

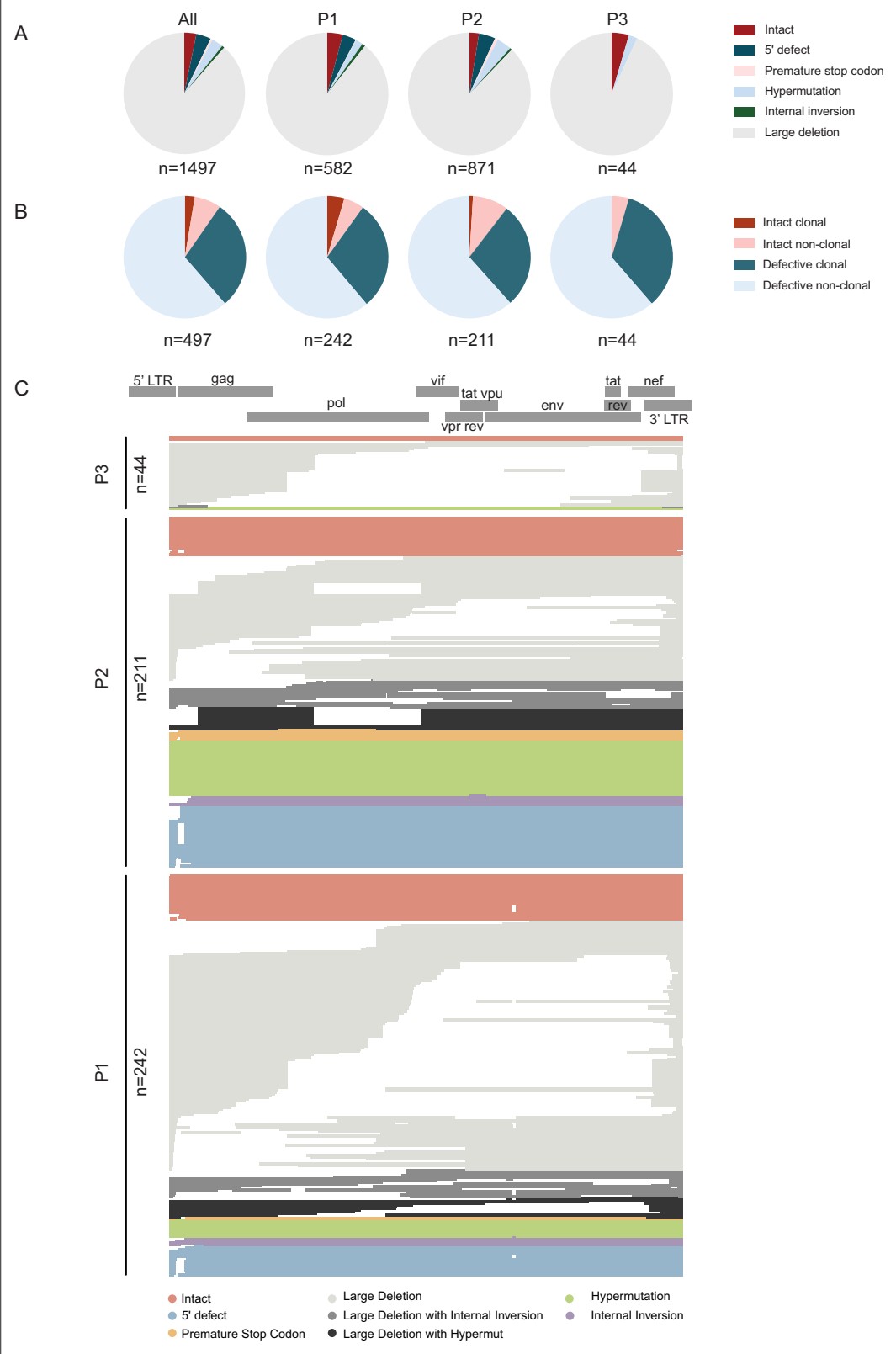

**Figure 1.** Proviral sequence classification in all analyzed HIV-1-infected cells from three study participants. (**A**) Pie charts reflecting proportions of proviruses classified as intact or defective in study participants 1-3 (P1-P3). All proviruses identified by single-genome, near-full-length, next-generation sequencing and by counting amplification products in agarose gel electrophoresis were included. The total number of individual sequences

*Figure 1 continued on next page*

*Figure 1 continued*

included is listed below each pie chart. (**B**) Pie charts indicating proportions of intact and defective proviruses detected once (classified as non-clonal) and detected multiple times (classified as clonal). The total number of proviral sequences identified by single-genome, near-full-length, next-generation sequencing is listed below each pie chart. (**C**) Virograms summarizing individual HIV-1 proviral sequences aligned to the HXB2 reference genome from each participant; color coding reflects the classification of proviral sequences.

sequences was identified; both of these clonal sequences were located in the prostate (*Figure 3A–B*). Fourteen clones of defective proviruses were observed across multiple tissues in participant 2 (*Figure 3C*).

Clonal proviral sequences in CNS tissues were detected in all three study participants. In study participant 1, members of a clone of defective proviruses were observed in the basal ganglia and the frontal lobe (*Figure 4A–B*). In participant 2, two clones of defective proviruses were noted in the thalamus, one clone was detected in cells from the basal ganglia, and members of a fourth clone were detected in the thalamus and the occipital lobe (*Figure 4C–D*), demonstrating rather extensive evidence for clonal proliferation of virally-infected cells in the brain. Two of the 44 proviral sequences detected in the central nervous system of participant 3 met the criteria for genome-intactness; one of those was isolated from the basal ganglia and one from periventricular white matter. Defective proviruses, most of them harboring large deletions, were detected in all five CNS tissues from participant 3 (*Figure 4E–F*). One large clone involving 9 defective proviral sequences in participant 3 was broadly distributed across different brain tissues, encompassing sequences in the occipital lobe, basal ganglia, thalamus, and periventricular white matter. The other clone of defective proviruses was only detected in the periventricular white matter. Again, these data suggest that clonal proliferation is a rather common feature of HIV-1 reservoir cells in the CNS.

## Viral tropism and immune selection footprints

Viral tropism was evaluated based on the env V3 region of the proviruses using the Geno2pheno algorithm. Notably, all proviral sequences containing the env V3 region of participant 1 (n=130, 53.7%) and 3 (n=6, 13.6%) were predicted to be CCR5-tropic (*Figure 5A*). In participant 2, 62.1% of proviral sequences (n=131) were predicted to be likely CXCR4-tropic, while 18.5% (n=39) were classified as CCR5-tropic; the remaining 19.4% (n=41) were classified as undetermined due to the lack of env V3 regions in this study participant (*Figure 5A*). Notably, approximately half of the 131 proviruses with predicted CXCR4 tropism from study participant 2 (n=66, 50.4%) were isolated from the prostate, followed by the spleen (n=38, 29.0%), lymph node (n=20, 15.3%) and thyroid gland (n=3, 2.3%) (*Figure 5B–C*). Among all proviruses (n=16) from CNS tissues of participant 2, only one provirus with a large deletion, isolated from the occipital lobe, had predicted CXCR4-tropism; the tropism of other proviruses from the CNS was unknown due to the lack of env V3 regions. As an additional analysis step, we evaluated footprints of immune selection pressure and mutations resulting in resistance to antiretroviral agents in intact proviral sequences from our study subjects. We noted that the frequency of viral amino acid residues associated with resistance to broadly neutralizing antibodies did not notably differ among sequences isolated from different tissue compartments (*Figure 5—figure supplement 1*). We did not observe sequence variations consistent with escape from antiretroviral agents in any of the intact proviral sequences analyzed here.

## Discussion

The lifelong persistence of viral reservoir cells makes HIV-1 infection an incurable disease that necessitates indefinite antiretroviral suppression therapy. However, the location of HIV-1 viral reservoir cells across different tissues has been difficult to assess in the past, due to the limited availability of tissue samples. Recent studies, pioneered by investigators of the 'Last Gift Cohort', have catalyzed investigations to characterize HIV-1 sequences in diverse organ systems, specifically in the CNS (*Tang et al., 2023*; *Chaillon et al., 2020*). In our study, we used single-genome near full-length proviral sequencing to evaluate the distribution of HIV-1 reservoir cells in multiple anatomical compartments from autopsy samples of three individuals living with HIV-1 and receiving antiretroviral therapy until the time of their decease. Consistent with previous studies (*Banga et al., 2016*; *Kuo et al., 2020*), intact proviruses

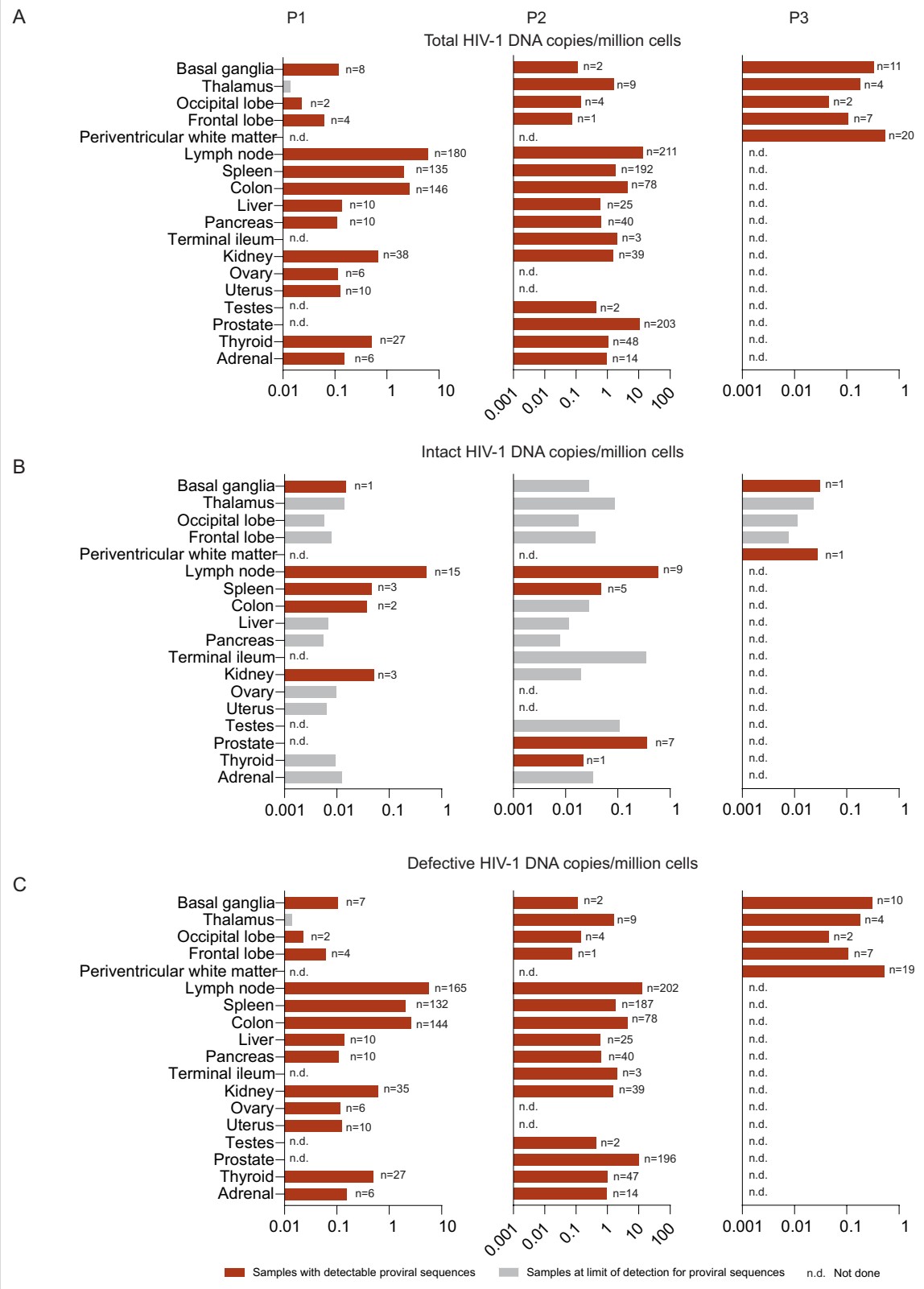

**Figure 2.** Distribution of total, intact, and defective HIV-1 proviruses in individual tissue compartments. Bar diagrams reflect relative frequencies of total (**A**), intact (**B**), and defective (**C**) proviruses in all analyzed tissues in study participants 1–3. The total number of individual proviral sequences determined by single-genome, near-full-length, next-generation sequencing and by counting amplification products in agarose gel electrophoresis from each tissue site of each participant is listed aside each bar. The red bars reflect samples with detectable proviral sequences; grey bars reflect samples at limit of

*Figure 2 continued on next page*

*Figure 2 continued*

detection for proviral sequences, calculated as 0.5 (single genome near–full-length PCR) copies per maximum number of cells tested without target identification (see Materials and Methods for details). N.d. (not done) indicates that the samples were not available from the indicated tissue sites.

The online version of this article includes the following figure supplement(s) for figure 2:

**Figure supplement 1.** Ratio of intact to defective proviral sequences in individual tissue compartments.

were readily detected in lymph nodes; moreover, we detected relatively high frequencies of intact proviruses in the colon, likely reflecting viral infection of CD4 T cells residing in gut-associated lymphoid tissues (GALT). Of note, our study is among the first investigations to identify near full-length proviral sequences from the central nervous system in two study participants, supporting the hypothesis that CNS cells can serve as reservoirs for long-term HIV-1 persistence despite antiretroviral therapy. Importantly, we noted that large clones of virally infected cells were broadly disseminated across multiple tissues, and, in selected cases, involved cells from the CNS; this suggests that HIV-1 reservoir cells seeded to the brain via hematogenous spread can proliferate in the local tissue microenvironment of the central nervous system. Together, our work suggests that HIV-1 reservoir cells harboring intact proviruses are broadly distributed across multiple anatomical locations, involving lymphoid tissues, gastrointestinal tissues, genitourinary tissues, and, importantly, central nervous system tissues.

HIV-1 can invade the CNS within days after infection, as demonstrated in animal (*Chakrabarti et al., 1991*) and in human studies (*Valcour et al., 2012*). Most likely, infection of CNS cells occurs as a result of transmigration of infected CD4 T cells and, possibly, macrophages across the blood-brain-barrier (BBB; *Spudich and González-Scarano, 2012*; *Liu et al., 2000*), a process that may be facilitated by the increased permeability of the BBB during the initial, highly replicative stage of HIV-1 infection. Infected cells that successfully enter the CNS may frequently be short-lived, however some infected CD4 T cells in the brain may persist long-term. Moreover, invading infected cells can transmit the virus to resident CNS cells via effective cell-to-cell transmission (*Liu et al., 2000*). At least three different CNS cell types seem to be susceptible to HIV-1 infection: perivascular macrophages, microglial cells, and astrocytes, although the role of the latter as HIV-1 target cells is more controversial (*Churchill et al., 2006*; *Wallet et al., 2019*; *Woodburn et al., 2022*). Yet, due to the difficulties in accessing brain tissues for analytic purposes, the role of the central nervous system in HIV-1 persistence during antiretroviral therapy remained largely unknown for a long time. Using the intact proviral DNA assay (IPDA), a ddPCR-based technique allowing to identify proviruses with a high probability of being genome-intact, previous investigators identified intact proviruses in 6 out of 9 ART-treated persons (*Cochrane et al., 2022*), although the precise proviral DNA sequence and their possible clonality was not assessed with this technology. In our study, a total of 13 CNS tissue samples from three study participants were analyzed, including specimens from the basal ganglia, thalamus, occipital lobe, frontal lobe, and periventricular white matter. In two study persons (participants 1 and 3), intact proviral sequences were detected in basal ganglia, suggesting that HIV-1 may preferentially persist in this anatomical compartment in the CNS; an additional intact provirus was detected in periventricular white matter. Notably, the intact provirus from basal ganglia in one of our study persons (participant 1) was clonal with 4 intact proviruses from the lymph node, indicating, to our knowledge for the first time, that CNS tissue can be involved in the multi-compartment dissemination of large clones of HIV-1 proviruses in ART-treated persons. Moreover, in multiple instances, we observed clones of HIV-1-infected cells that were distributed across different autologous CNS tissues, specifically in study participant 3, suggesting local spread of virally infected cells through clonal proliferation within the immune microenvironment of the CNS.

Our study did not allow to determine which cell types were infected by HIV-1 and responsible for clonal expansion of viral reservoir cells in the CNS; however, it is possible that infected microglia are involved. Microglial cells originate from erythromyeloid progenitors in the yolk sac and colonize the developing CNS during embryogenesis (*Kierdorf et al., 2013*); these cells act as the main innate immune cell population of the CNS. Due to their long half-life (typically several years), their ability to divide and self-renew, and their high cell-intrinsic susceptibility to HIV-1 (*Cenker et al., 2017*), these cells may represent a primary cellular site for long-term HIV-1 persistence in the CNS. In particular, self-renewal through homeostatic proliferation in microglia (*Réu et al., 2017*) may support HIV-1 persistence through clonal expansion. A recent study indeed identified HIV-1 DNA and RNA in microglia cells from autopsies from ART-treated PLWH who did not have specific (HIV or non-HIV

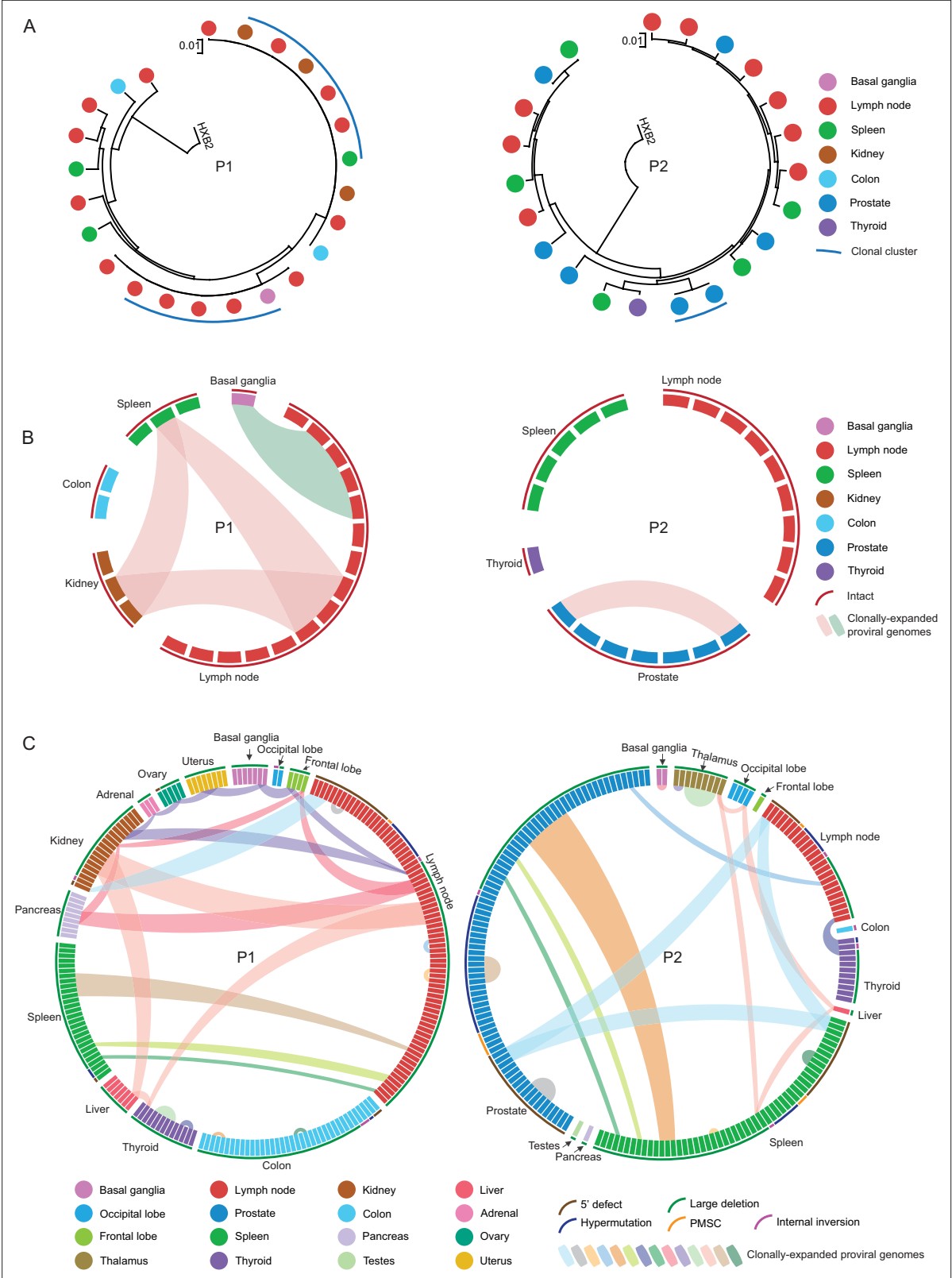

**Figure 3.** Dissemination of HIV-1-infected cells across multiple anatomical tissues in participants 1 and 2. (**A and B**) Circular maximum likelihood phylogenetic trees (**A**) and circos plots (**B**) of intact proviral sequences from participant 1 (P1) and participant 2 (P2). HXB2, reference HIV-1 sequence. Color coding reflects tissue origins. Each symbol reflects one intact provirus. Clonal intact sequences, defined by complete sequence identity, are indicated by blue arches in (**A**) and by internal connections in (**B**). (**C**) Circos plots reflecting the clonality of defective proviral sequences from participant

*Figure 3 continued on next page*

Figure 3 continued

1 (left panel) and participant 2 (right panel). Each symbol reflects one defective provirus. Clonal sequences, defined by complete sequence identity, are highlighted. Color-coded arches around the plots indicate types of defects in HIV-1 genomes.

associated) CNS pathology (**Ko et al., 2019**). Evidence for HIV-1 persistence in CD68 +myeloid cells, most likely microglia cells, was also described by previous investigators (**Cochrane et al., 2022**). That said, the presence of clonal proviral sequences shared between the CNS and lymphoid tissues in our study could also suggest that migrating CD4 T cells infected with R5-tropic viruses may infect the brain as 'Trojan horses', and then potentially clonally expand in situ in the CNS; this hypothesis is consistent with recent findings from **Kincer et al., 2023**. In the future, it may be possible to capture the phenotypic characteristics of HIV-1 reservoir cells from the CNS directly with single-cell assays that permit combined assessments of the phenotype and the proviral sequence; an example for such an assay system was recently described (**Sun et al., 2023**).

In our study, CXCR4-tropic proviruses were exclusively detected in participant 2. Compared to the other two participants who began ART shortly after HIV-1 diagnosis, participant 2 was diagnosed with HIV-1 25 years prior to starting antiretroviral therapy and died 10 months after ART commencement; therefore, viral CXCR4 tropism most likely resulted from 'coreceptor switch' frequently occurring during advanced stages of immune deficiency (**Connor et al., 1997**). Whether the preferential persistence of CXCR4-tropic viruses in study participant 2 was associated with our inability to detect intact proviruses in the CNS in this person is unclear; however, prior studies suggested that CCR5 can act as the principal co-receptor for HIV-1 isolates in the brain (**Albright et al., 1999**; **He et al., 1997**). Moreover, rebound viremia in cerebrospinal fluid after ART interruption is mostly fueled by CCR5-tropic virus (**Kincer et al., 2023**), further supporting the assumption that R5-tropic viruses are better adjusted to persist in the brain. Another notable finding in our study was the high number of HIV-1 proviruses isolated from the prostate, which had the second highest proviral frequency among all analyzed tissues in study participant 2, second only to lymph node samples. Other studies also reported that the prostate can represent a tissue reservoir for HIV-1 (**Chaillon et al., 2020**). We were unable to identify the precise cell type harboring HIV-1 in the prostate, but it is possible that myeloid cells may harbor HIV-1 in this location. A prior study indeed demonstrated that intact, replication-competent HIV-1 can persist in myeloid cells from the urethra, located in immediate anatomical proximity to the prostate (**Ganor et al., 2019**).

Our study has several limitations. Importantly, this work includes only 3 participants, and brain tissues were the only samples available from participant 3. Moreover, due to limited tissue sizes available for investigation, very few cells were assayed from the terminal ileum and testes; prior studies suggested high HIV-1 DNA levels in these 2 tissues during suppressive antiretroviral therapy (**Horn et al., 2021**; **Miller et al., 2019**). Another limitation was that peripheral blood samples were not available from the 3 participants, which made it impossible to study phylogenetic associations between tissue reservoirs of HIV-1 relative to viral species circulating in peripheral blood. Moreover, we cannot fully exclude contamination of tissue samples with cells from peripheral blood. However, after autopsy, the tissue samples were washed thoroughly with PBS to eliminate blood as much as possible. Notably, we failed to detect intact proviral sequences from over 100 million liver cells of 2 participants, despite the fact that the liver contains about 13% of all human blood supply at any given time point, arguing against contamination of tissues with blood cells.

In sum, this study provides a deep analysis of tissue reservoirs for HIV-1 that includes a detailed assessment of HIV-1 sequences in CNS tissues. Our work supports the persistence of genome-intact HIV-1 in many tissues, including CNS tissues, emphasizing the difficulties in finding strategies to effectively eliminate HIV-1 from the human body in clinical settings.

## Materials and methods
### Study Participants

HIV-1-infected study participants were recruited at the Massachusetts General Hospital (MGH) and the Brigham and Women's Hospital in Boston, MA. Fresh tissues were sampled during routine autopsy according to protocols approved by the Institutional Review Board and cryopreserved for future study according to standard protocols. The clinical characteristics of study participants are summarized in

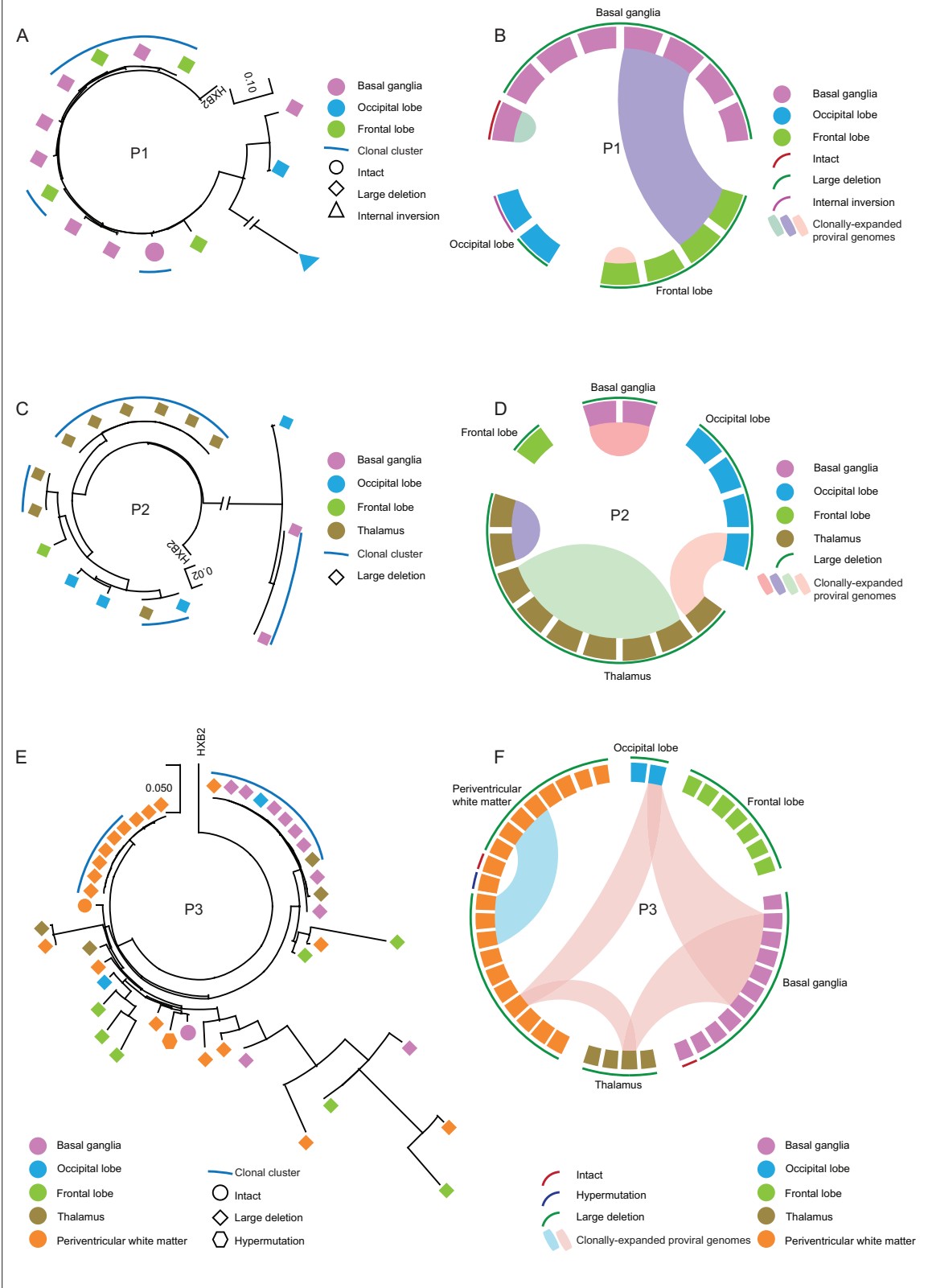

**Figure 4.** Dissemination of HIV-1-infected cells across CNS tissues. (**A, C, and E**) Circular maximum likelihood phylogenetic trees of all proviral sequences derived from CNS tissues of the three study participants (**A**, participant 1; **C**, participant 2; **E**, participant 3). Color coding reflects tissue origins. Clonal sequences, defined by complete sequence identity, are indicated by blue arches. (**B, D, and F**) Circos plot reflecting the clonality of all proviral sequences isolated from CNS tissues of three participants (**B**, participant1; **D**, participant 2; **F**, participant 3). Each symbol reflects one provirus. Clonal sequences, defined by complete sequence identity, are highlighted. Color-coded arches around the plots indicate types of proviral sequences.

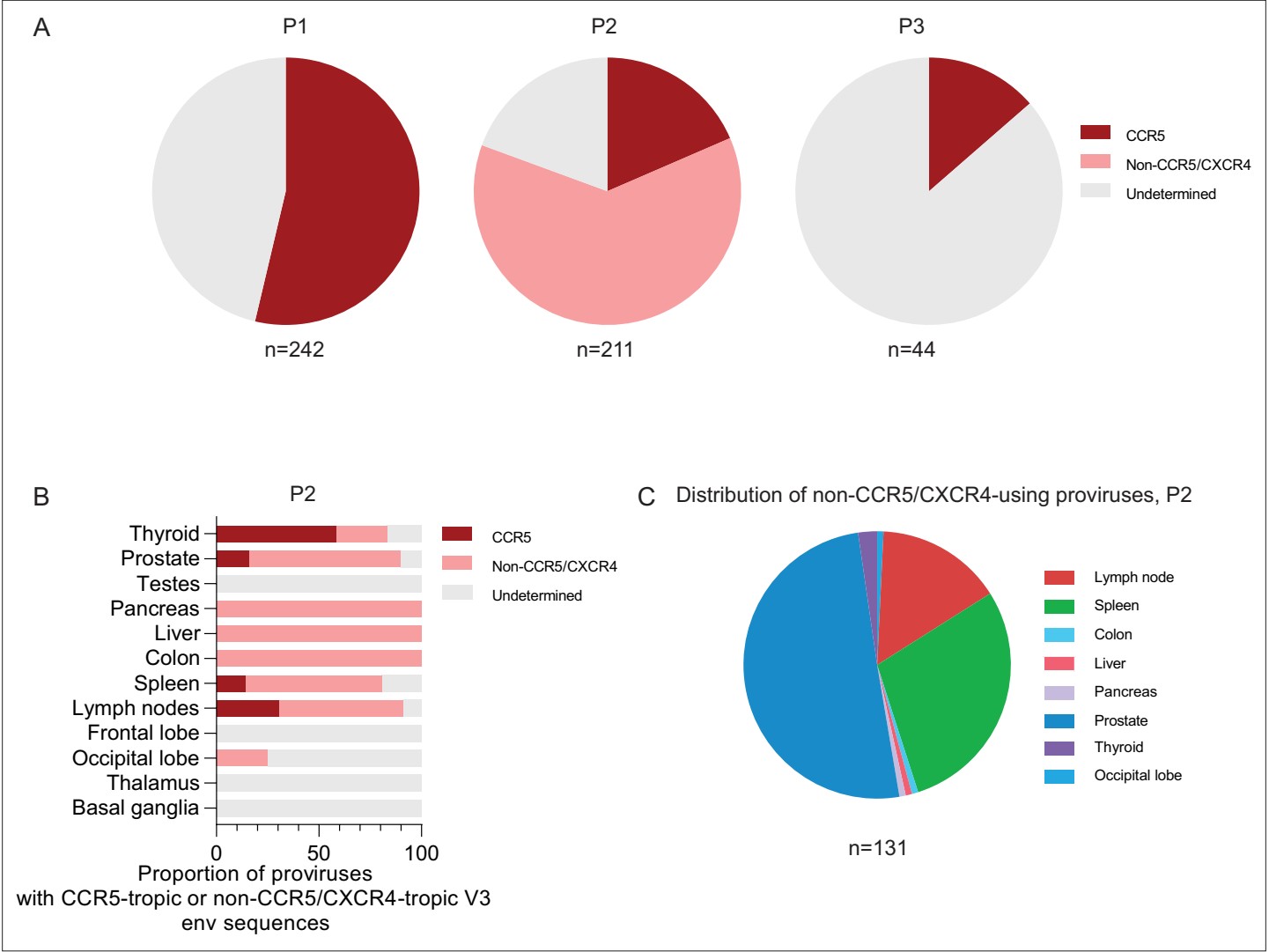

**Figure 5.** HIV-1 tropism analysis of proviral sequences. (**A**) Pie charts indicating the proportions of all proviruses from each participant with CCR5-tropic or non–CCR5/CXCR4-tropic V3 envelope sequences are shown. The total number of proviral sequences included in this analysis is listed below each pie chart. (**B**) The proportions of proviruses with CCR5-tropic or non-CCR5/CXCR4-tropic V3 env sequences in each tissue from participant 2 are shown. (**C**) Pie charts indicating the proportions of all non–CCR5-tropic proviruses from participant 2 are shown. Color coding reflects tissue origins. The total number of analyzed proviral sequences is listed below the pie chart. HIV-1 tropism was computationally inferred using Geno2pheno (https://coreceptor.geno2pheno.org/). HIV-1 tropism was classified as "CCR5" if the false-positive rate (FPR) predicted by Geno2pheno was >2% and 'CXCR4' if FPR was <2%.

The online version of this article includes the following figure supplement(s) for figure 5:

**Figure supplement 1.** Predicted susceptibility of intact proviruses to broadly neutralizing antibodies (bnAbs).

---

*Table 1*. All study participants gave written consent to donate their tissues for autopsy studies, and the study was conducting according to the declaration of Helsinki.

## HIV-1 DNA quantification by IPDA

Tissue samples were dissected and subjected to genomic DNA extraction using the DNeasy Blood and Tissue Kit (QIAGEN DNeasy, #69504). HIV-1 DNA was analyzed by the Intact Proviral DNA Assay (IPDA), using primers and probes described previously (*Bruner et al., 2019*). PCR was performed using the following program: 95 °C for 10 min, 45 cycles of 94 °C for 30 s and 59 °C for 1 min, 98 °C for 10 min. The droplets were subsequently read by the QX200 droplet reader (Bio-Rad), and data were analyzed using QuantaSoft software (Bio-Rad).

## Near full-length HIV proviral sequencing

Genomic DNA diluted to single HIV-1 genome levels based on Poisson distribution statistics and IPDA results was subjected to near full-genome HIV-1 amplification using a one-amplicon approach (*Lee et al., 2017*; *Lee et al., 2019*). PCR products were visualized by agarose gel electrophoresis. Amplification products were individually subjected to Illumina MiSeq sequencing at the MGH DNA Core facility. The resulting short reads were de novo assembled using Ultracycler v1.0 and aligned to HXB2 to identify large deleterious deletions (<8000 bp of the amplicon aligned to HXB2), out-of-frame indels, premature/lethal stop codons, internal inversions, or 5′-LTR defect (≥15 bp insertions and/or deletions relative to HXB2), using an automated in-house pipeline written in Python scripting language (https://github.com/BWH-Lichterfeld-Lab/Intactness-Pipeline; *Gao et al., 2020*). The presence/absence of APOBEC-3G/3 F–associated hypermutations were determined using the Los Alamos HIV Sequence Database Hypermut 2.0 program. The sequences of individual genes were extracted by GeneCutter, and the start codons of Gag, Pol, and Env were examined and considered. Viral sequences that lacked all defects listed above were classified as "genome-intact". Multiple sequence alignments were performed using MUSCLE (*Edgar, 2004*). Phylogenetic analyses were conducted using MEGA X, applying maximum likelihood approaches. Viral sequences were considered clonal if they had completely identical consensus sequences; single-nucleotide variations in primer binding sites were excluded for clonality analysis. When viral DNA sequences were undetectable, data were reported as LOD (limit of detection), calculated as 0.5 copies per maximum number of cells tested without target identification. The sensitivity of proviral species to broadly-neutralizing antibodies (bnAb) was estimated by calculating the number of amino acid signature sites associated with sensitivity to four bnAb classes within the env amino acid sequence from each intact provirus, as previously described (*Bricault et al., 2019*).

## Data analysis and statistics

Data are summarized as bar graphs. Phylogenetic relationships were evaluated using maximum-likelihood phylogenetic trees. Images were prepared using Adobe Illustrator. HIV-1 tropism was computationally inferred using Geno2pheno (https://coreceptor.geno2pheno.org/). HIV-1 tropism was classified as 'CCR5' if the false-positive rate (FPR) predicted by Geno2pheno was >2%, however, 92% of our proviral sequences meeting this definition had a FPR score >10%; proviruses were considered 'CXCR4-tropic' if FPR was <2%.

## Acknowledgements

ML is supported by NIH grants AI117841, AI120008, AI130005, DK120387, AI152979, AI155233, AI135940 and by the American Foundation for AIDS Research (amfAR, #110181–69-RGCV). XGY is supported by NIH grants AI155171, AI116228, AI078799, MH134823, HL134539, DA047034, amfAR ARCHE Grant # 110393–72-RPRL and the Bill and Melinda Gates Foundation (INV-002703). ML and XGY are members of the DARE, ERASE, PAVE and BEAT-HIV Martin Delaney Collaboratories (UM1 AI164560, AI164562, AI164566, AI164570). IHS is supported by NIH grant R21NS119660.

## Additional information

### Funding

| Funder | Grant reference number | Author |
|---|---|---|
| National Institutes of Health | AI117841 | Mathias Lichterfeld |
| National Institutes of Health | AI120008 | Mathias Lichterfeld |
| National Institutes of Health | AI130005 | Mathias Lichterfeld |
| National Institutes of Health | DK120387 | Mathias Lichterfeld |

| Funder | Grant reference number | Author |
| --- | --- | --- |
| National Institutes of Health | AI152979 | Mathias Lichterfeld |
| National Institutes of Health | AI155233 | Mathias Lichterfeld |
| National Institutes of Health | AI135940 | Mathias Lichterfeld |
| American Foundation for AIDS Research | amfAR,#110181-69-RGCV | Mathias Lichterfeld |
| National Institutes of Health | AI155171 | Xu G Yu |
| National Institutes of Health | AI116228 | Xu G Yu |
| National Institutes of Health | AI078799 | Xu G Yu |
| National Institutes of Health | MH134823 | Xu G Yu |
| National Institutes of Health | HL134539 | Xu G Yu |
| National Institutes of Health | DA047034 | Xu G Yu |
| amfAR ARCHE | Grant # 110393-72-RPRL | Xu G Yu |
| Bill and Melinda Gates Foundation | INV-002703 | Xu G Yu |
| DARE, ERASE, PAVE and BEAT-HIV Martin Delaney Collaboratories | UM1 AI164560,AI164562,AI 164566,AI164570 | Xu G Yu Mathias Lichterfeld |
| National Institutes of Health | R21NS119660 | Isaac H Solomon |

The funders had no role in study design, data collection and interpretation, or the decision to submit the work for publication.

## Author contributions

Weiwei Sun, Formal analysis, Investigation, Methodology, Writing – original draft; Yelizaveta Rassadkina, Methodology; Ce Gao, Data curation, Formal analysis, Methodology; Sarah Isabel Collens, Isaac H Solomon, Resources; Xiaodong Lian, Formal analysis; Shibani S Mukerji, Conceptualization, Resources, Investigation; Xu G Yu, Conceptualization, Formal analysis, Supervision, Funding acquisition, Project administration; Mathias Lichterfeld, Conceptualization, Formal analysis, Supervision, Funding acquisition, Investigation, Methodology, Writing – original draft, Project administration, Writing - review and editing

## Author ORCIDs

Weiwei Sun ORCID https://orcid.org/0000-0002-4578-6292
Mathias Lichterfeld ORCID http://orcid.org/0000-0001-9865-8350

## Ethics

Human subjects: Tissues were sampled during routine autopsy according to protocols approved by the Institutional Review Board of MassGeneralBrigham (MGB) and cryopreserved for future study according to standard protocols.

Reviewer #1 (Public Review): https://doi.org/10.7554/eLife.89837.3.sa1
Reviewer #2 (Public Review): https://doi.org/10.7554/eLife.89837.3.sa2
Author Response https://doi.org/10.7554/eLife.89837.3.sa3

## Additional files

### Supplementary files
• MDAR checklist

### Data availability
Proviral sequencing data have been deposited in Genbank (accession numbers: OR660701–OR661197) and may be accessed at https://www.ncbi.nlm.nih.gov/nuccore.

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
