## [Editor Report · eLife assessment]

This **important** study uses near full-length HIV-1 sequencing to examine proviral persistence in various tissues derived from three individuals who received antiretroviral therapy until time of death. Intact as well as defective HIV-1 proviruses are found at various anatomical sites including the central nervous system; the results are **convincing** and relevant for our understanding of latent viral reservoirs, especially in the brain.

---

## [Referee Report · Reviewer #1 (Public Review)]

Despite durable viral suppression by antiretroviral therapy (ART), HIV-1 persists in cellular reservoirs in vivo. The viral reservoir in circulating memory T cells has been well characterized, in part due to the ability to safely obtain blood via peripheral phlebotomy from people living with HIV-1 infection (PWH). Tissue reservoirs in PWH are more difficult to sample and are less well understood. In this small (n=3) autopsy study, Sun and colleagues use an advanced genetic sequencing technique to characterize HIV-1 that persists in human tissues despite antiretroviral therapy. The authors describe isolation and genetic characterization of HIV-1 reservoirs from a variety of tissues including the central nervous system (CNS) obtained from three recently deceased individuals at autopsy. They identified clonally expanded proviruses in the CNS in all three individuals.

Strengths of the work include the study of human tissues that are under-studied and difficult to access, and the sophisticated near-full length sequencing technique that allows for inferences about genetic intactness and clonality of proviruses. The small sample size (n=3) is a drawback. Furthermore, two individuals were on ART for just one year at the time of autopsy and had T cells compatible with AIDS, and one of these individuals had a low-level detectable viral load (Figure S1). This makes generalizability of these results to PWH who have been on ART for years or decades and have achieved durable viral suppression and immune reconstitution difficult.

While anatomic tissue compartment and CNS region accompany these PCR results, it is unclear which cell types these viruses persist in. As the authors point out, it is possible that these reservoir cells might have been infiltrating T cells from blood present at the time of autopsy tissue sampling. Cell type identification would greatly enhance the impact of this work. Overall, this small, thoughtful study contributes to our understanding of the tissue distribution of persistent HIV-1, and informs the ongoing search for viral eradication.

---

## [Referee Report · Reviewer #2 (Public Review)]

The authors were trying to survey reservoir viral sequences in different anatomical sites in the body, with the brain being of special interest. This is a study that is technically demanding and here is well done, providing insights that prompt new and more sophisticated questions.

The authors use end-point dilution PCR to identify individual proviruses that can then be sequenced with high accuracy. These are high quality data sets but given the technical requirements of this approach the number of sequenced proviruses is limiting given the scope of questions this study addresses. Nonetheless, there is a lot of data here to draw many useful conclusions.

It will be important to realize how clones of infected T cells can move around the body, including into the CNS compartment. It will also be important to remember that there are limits in sampling depth of proviruses in any one tissue meaning the failure to detect something has a limit in sensitivity of detection when trying to interpret a negative result.

As noted in the next section, it is important to emphasize that there is another entry phenotype beyond X4 that will ultimately be important in interpreting these results. Macrophage-tropic viruses are often found in the CNS compartment and it will be important to understand whether these CNS-derived sequences are macrophage-tropic viruses there infecting macrophages and microglia or if they are all T-tropic viruses brought in by wandering infected T cells (or both).

---

## [Author Response]

The following is the authors’ response to the original reviews.

We are very grateful to the reviewers for their insightful and detailed analysis of our work, in particular to reviewer 2. We also would like to thank the Elife editorial team for organizing this form of public review and debate, which we believe will be of interest to the science community.

**Reviewer #1 (Public Review):**
Despite durable viral suppression by antiretroviral therapy (ART), HIV-1 persists in cellular reservoirs in vivo. The viral reservoir in circulating memory T cells has been well characterized, in part due to the ability to safely obtain blood via peripheral phlebotomy from people living with HIV-1 infection (PWH). Tissue reservoirs in PWH are more difficult to sample and are less well understood. Sun and colleagues describe isolation and genetic characterization of HIV-1 reservoirs from a variety of tissues including the central nervous system (CNS) obtained from three recently deceased individuals at autopsy. They identified clonally expanded proviruses in the CNS in all three individuals.Strengths of the work include the study of human tissues that are under-studied and difficult to access, and the sophisticated near-full length sequencing technique that allows for inferences about genetic intactness and clonality of proviruses. The small sample size (n=3) is a drawback. Furthermore, two individuals were on ART for just one year at the time of autopsy and had T cells compatible with AIDS, and one of these individuals had a low-level detectable viral load (Figure S1). This makes generalizability of these results to PWH who have been on ART for years or decades and have achieved durable viral suppression and immune reconstitution difficult.While anatomic tissue compartment and CNS region accompany these PCR results, it is unclear which cell types these viruses persist in. As the authors point out, it is possible that these reservoir cells might have been infiltrating T cells from blood present at the time of autopsy tissue sampling. Cell type identification would greatly enhance the impact of this work. Several other groups have undergone similar studies (with similar results) using autopsy samples (links below). These studies included more individuals, but did not make use of the near-full length sequencing described here. In particular, the Last Gift cohort, based at UCSD and led by Sara Gianella and Davey Smith, has established protocols for tissue sampling duringautopsy performed soon after death. https://pubmed.ncbi.nlm.nih.gov/35867351/
https://pubmed.ncbi.nlm.nih.gov/37184401/

We agree with reviewer 1 that studies to identify specific cell types that harbor intact HIV-1 in individual tissue compartments would be very informative; our group has recently initiated such studies.

Overall, this small, thoughtful study contributes to our understanding of the tissue distribution of persistent HIV-1, and informs the ongoing search for viral eradication.

We thank reviewer 1 for these encouraging remarks.

**Reviewer #2 (Public Review):**
The manuscript by Sun et al. applies the powerful technology of profiling viral DNA sequences in numerous anatomical sites in autopsy samples from participants who maintained their antiviral therapy up to the time of death. The sequencing is of high quality in using end-point dilution PCR to generate individual viral genomes. There is a thoughtful discussion, although there are points that we disagree with. This is an important data set that increases the scope of how the field thinks about the latent reservoir with a new look at the potential of a reservoir within the CNS.

We greatly appreciate the comments by reviewer 2 and would like to thank them for their detailed and very knowledgeable analysis of this paper.

1. The participants are very different in their exposure to HIV replication and disease progression. Participant 1 appears to have been on ART for most of the time after diagnosis of infection (16 years) and died with a high CD4 T cell count. The other two participants had only one year on ART and died with relatively low CD4 T cell counts (under 200). This could lead to differences in the nature of the reservoir. In this regard, the amount of DNA per million cells appears to be about 10-fold lower across the compartments sampled for participant 1. Also, one might expect fewer intact proviruses surviving after 16 years on ART compared to only 1 year on ART. The depth of sampling may be too limited and the number of participants too few to assess if these differences are features of these participants because of their different exposures to HIV replication. On the positive side, finding similarities across these big differences in participant profiles does reinforce the generalizability of the observations.

Many thanks for pointing this out. We also noticed that the total number of HIV-1 proviruses is smaller in our study participant 1 (who had been on ART for 16 years), compared to study persons 2 and 3 with more limited treatment durations (1-2 years), however, due to the small number of study persons, we think we cannot use these results for inferring how treatment duration influences viral reservoir size in tissues.

1. The following analysis will be limited by sampling depth but where possible it would be interesting to compare the ratio of intact to defective DNA. A sanctuary might allow greater persistence of cells with intact viral DNA even without viral replication (i.e. reduced immune surveillance). Detecting one or two intact proviruses in a tissue sample does not lend itself to a level of precision to address this question, but statistical tests could be applied to infer when there is sampling of 5 or more intact proviruses to determine if their frequency as a ratio of total DNA in different anatomical sites is similar or different. This would allow adjustment for the different amount of viral DNA in different compartments while addressing the question of the frequency of intact versus defective proviruses. One complication in this analysis is if there was clonal expansion of a cell with an intact genome which would represent a fortuitous overrepresentation intact genomes in that compartment.

We have performed the analysis suggested by reviewer 2 and included a diagram reflecting the ratio of intact/defective proviruses as a new supplemental figure (Figure S2). Unfortunately, we do not feel comfortable to draw any real conclusions from this additional analysis; the sample sizes are simply too limited.

1. The key point of this work is that the participants were on therapy up to the time of death ("enforcing" viral latency). The predominance of defective genomes is consistent with this assumption. Is there data from untreated infections to compare to as a signature of whether the viral DNA population was under selective pressure from therapy or not? Presumably untreated infections contain more intact DNA relative to total DNA. This would represent independent evidence that therapy was in place.

We agree that an analysis of autopsy samples from untreated persons living with HIV-1 would be of great interest, and are actively collaborating with neuropathologists from multiple sites to obtain such samples. Yet, we are not convinced that selection pressure on reservoir cells during ART can be appropriately identified through quantitative virological assays. Rather, we feel that the selection of proviruses can be best assessed when qualitative parameters, including proviral integration sites and their position relative to host epigenetic chromatin features, are evaluated.

1. There are several points in Figure 5 to raise about V3 loop sequences. The analysis includes a large number of "undetermined" sequences that did not have a V3 loop sequence to evaluate. We would argue it is a fair assumption that the deleted proviruses have the same distribution of X4 and R5 sequences as the ones that have a V3 sequence to evaluate. In this view it would be possible to exclude the sequences for which there is no data and just look at the ratio of X4 and R5 in the different compartments, specifically does this ratio change in a statistically significant way in different compartments? The authors use "CCR5 and non-CCR5" as the two entry phenotypes. The evidence is pretty strong that the "other" coreceptor the virus routinely uses is CXCR4, and G2P is providing the FPR for X4 viruses. Perhaps the authors are trying to create some space for other coreceptors on microglia, but we are pretty sure what they are measuring is X4 viruses, especially in this late disease state of participant 2. Finally, we have previously observed that the G2P FPR score of <2 is a strong indicator of being X4, FPR scores between 2 and 10 have a 50% chance of being X4, and FPR scores above 10 are reliably R5 (PMID27226378). In addition, we observed that X4 viruses form distinct phylogenetic lineages. The authors might consider these features of X4 viruses in the evaluation of their sequences. Specifically, it would be helpful to incorporate the FPR scores of the reported X4 viruses.

Many thanks for these thoughts. We have now included FPR scores for all sequences and considered sequences with FPR score <2 as X4-tropic. Among 497 proviral sequences derived from all three participants, only 14 proviral sequences had FPR scores between 2 and 10 and their tropism was classified as CCR5 in the new Figure 5. We agree that viral tropism analysis of proviral sequences from the CNS would be of particular interest for study subject 2; however, most brain-derived sequences from that person had large deletions in the env region, precluding an analysis of viral tropism.

1. We have puzzled over the many reports of different cell types in the CNS being infected. When we examined these cell types (both as primary cells and as iPSC-derived cells), all cells could be infected with a version of HIV that had the promiscuous VSV-G protein on the virus surface as a pseudotype. However, only macrophages and microglia could be infected using the HIV Env protein, and then only if it was the M-tropic version and not the T-tropic version (PMID35975998). RNAseq analysis was consistent with this biological readout in that only macrophages and microglia expressed CD4, neurons and astrocytes do not. From the virology point of view, astrocytes are no more infectable than neurons.

We appreciate these comments. As described in our discussion, we agree that the role of astrocytes as target cells for HIV-1 infection is highly controversial; we look forward to future opportunities to evaluate HIV sequences in sorted astrocytes from autopsy tissues.

1. The brain gets exposed to virus from the earliest stages of infection but this is not synonymous with viral replication. Most of the time there is virus in the CSF but it is present at 1-10% of the level of viral load in the blood and phylogenetically it looks like the virus in the blood, most consistent with trafficking T cells, some of which are infected (PMID25811757). The fact that the virus in the blood is almost always T cell-tropic in needing a high density of CD4 for entry makes it unlikely that monocytes are infected (with their low density of CD4) and thus are not the source of virus found in the CNS. It seems much more likely that infected T cells are the "Trojan Horse" carrying virus into the CNS.

We appreciate the reviewer’s referral to Greek mythology and agree that the hypothesis of infected T cells acting as “Trojan horses” is more intuitive and better supported by available data. We have adjusted our discussion accordingly.

1. While all participants were taking antiretroviral therapy at the time of their death, they were not all suppressed when the tissues were collected. The authors are careful not to mention "suppressive ART" in the text, which is appreciated. However, the title should be changed to also reflect this fact.

Thanks for pointing this out. From our perspective, ART is never fully suppressive, as low-level viremia (below the detection threshold of commercial PCR assays) is detectable in almost all ART-treated persons. As such, it is not clear to us that “suppressive” necessarily implies suppression below the detection limits of commercial PCRs. We argue that ART can also be suppressive when plasma viral loads are in the range of 100 copies/ml, as they are in our study subject 3. Nevertheless, we have changed the title to avoid confusion.

**Reviewer #1 (Recommendations For The Authors):**
I encourage the authors to compare their autopsy and tissue sampling procedures to those used by The Last Gift researchers and consider including references to this ongoing study. If the authors plan to continue in this line of research, the field would greatly benefit from a collaboration that would bring together their excellent and advanced PCR technique with the larger sample size offered by The Last Gift. Lastly, is there some way to simultaneously determine cell type when NFL sequencing is performed?

We look forward to collaborating with investigators from the Last Gift Cohort in the future and have integrated additional references in the manuscript to acknowledge their work. At the current stage of technology development, we think that sorting of infected cells based on canonical markers of defined cell populations is the preferred approach for identifying phenotypic properties of infected cells; however, expansion of the PheP-Seq assay (Sun et al., Nature 2023), may facilitate this process in the future.

**Reviewer #2 (Recommendations For The Authors):**
1. The authors have chosen to lump all R5 viruses together in terms of their entry phenotype, giving all viruses an equal chance of infecting all potentially susceptible cell types. This ignores the fact that normal HIV is selected to infect cells, requiring a high density of CD4 as is found on T cells. We use the term R5 T cell-tropic to describe "normal" HIV. The ability to efficiently enter cells that have a low density of CD4, such as macrophages and microglia, involves the evolution of a distinct phenotype, termed macrophage tropism (PMID24307580, and work of others). This happens most often in the CNS where T cells are infrequent thus potentiating evolution to infect an alternative cell type. This change in entry phenotype is dramatic and, like X4 viruses, results in phylogentically distinct lineages (PMID22007152). There are no sequence signatures for M-tropic viruses as there are for X4 viruses, but the fact that there are sequences shared between the CNS and lymphoid tissue makes it much more likely that there are T cells migrating around the body, including into the CNS, that are carrying R5 T cell-tropic virus with them, with the cells potentially clonally expanding in situ in the CNS. The persistence of a potential CNS T cell reservoir was the point we were trying to make in our recent paper (ref. 38), not only that these CSF rebound viruses were R5 viruses but they were selected for replication in T cells as seen by their dependence of a high density of CD4 for entry. This is the conclusion one would reach if clonally expanded viral sequences were shared between two lymphoid compartments. It is not necessary to ascribe properties of infection and clonal amplification to microglia cells when a more parsimonious explanation is that there are low levels of T cells in the CNS, especially in the absence of entry phenotype data showing these sequences encode an M-tropic entry phenotype. As is the authors are just adding to the unproven belief that virus in the CNS must be in myeloid cells, which in this case in particular we suspect is the wrong interpretation.

We are impressed by reviewer 2’s recent work, suggesting the viral reservoir in the CNS may primarily consist of clonally-expanded R5 T-cell tropic viruses. We have adjusted our discussion to emphasize this possibility, and to highlight that viral entry phenotyping data will be informative for better understanding viral persistence in the brain.

1. The authors noted that the frequency of intact proviruses is highest in the lymph nodes of 2/2 participants for which they had lymph node samples, relative to the other tissues examined. They thus conclude, "Together, these results indicate that intact HIV-1 proviruses are preferentially detected in lymphoid and gastrointestinal (GI) tissues." However, an examination of Figure 2 reveals that the total HIV copy number is highest in the lymph nodes of these two people. Thus, it doesn't seem like HIV is preferentially intact in the lymph nodes as much as they sampled more provirus from that tissue and therefore were able to detect more intact proviruses.

We have adjusted our manuscript to indicate that the highest numbers of intact HIV-1 proviruses were present in lymph nodes, both in terms of absolute numbers and after normalization to the total numbers of cells analyzed.

1. In Figure 1A, the legend should be changed so that "PMSC" is spelled out as "premature stop codon" for ease of reading. This is done for Figure 1B.

We have corrected this issue as suggested by the reviewer.

1. The pie charts in Figure 5 could be better labeled for ease of interpreting. In Figure 5C, instead of just labeling it as "P2" it could be "Distribution of CXCR4-using proviruses, P2", as an example. As it stands, it is hard to know what the figure is describing without reading the text.

We have changed this accordingly.

1. While all participants were taking antiretroviral therapy at the time of their death, they were not all suppressed when the tissues were collected. The authors are careful not to mention "suppressive ART" in the text, which is appreciated. However, the title should be changed to also reflect this fact.

Thanks for pointing this out. From our perspective, ART is never fully suppressive, as low-level viremia (below the detection threshold of commercial PCR assays) is detectable in almost all ART-treated persons. As such, it is not clear to us that “suppressive” necessarily implies suppression below the detection limits of commercial PCRs. We argue that ART can also be suppressive when plasma viral loads are in the range of 100 copies/ml. Nevertheless, we have changed the title to avoid confusion.

**Editorial comments:**
In addition to the reviewers suggestion, we feel that adding more information on how you define intact proviral sequence, e.g. are only disrupted essential genes or also in accessory genes considered? Previous studies have shown that brain-derived HIV-1 strains are usually CCR5-tropic, show high affinity for the CD4 receptor and frequently contain defective vpu genes. Some information and discussion if the brainderived sequences confirm these previous finding seems of significant interest.

As described in our previous work (e. g. Lee et al, JCI 2017; Jiang et al, Nature 2020), accessory genes are not considered in our definition of “genome intactness”; this is consistent with approaches other investigators have chosen (e. g. Hiener et al, Cell Reports 2017). Within the genome intact sequences we identified in the CNS in our study persons, we found no evidence for deletions of vpu sequences; this has been emphasized in the revised manuscript.